# Neonatal Clonazepam Administration Induced Long-Lasting Changes in GABA_A_ and GABA_B_ Receptors

**DOI:** 10.3390/ijms21093184

**Published:** 2020-04-30

**Authors:** Hana Kubová, Zdeňka Bendová, Simona Moravcová, Dominika Pačesová, Luisa Rocha, Pavel Mareš

**Affiliations:** 1Institute of Physiology, Academy of Sciences of the Czech Republic, 14220 Prague, Czech Republic; maresp@biomed.cas.cz; 2Faculty of Science, Charles University, 12800 Prague, Czech Republic; zdenka.bendova@natur.cuni.cz (Z.B.); simona.moravcova@natur.cuni.cz (S.M.); dominika.pacesova@natur.cuni.cz (D.P.); 3National Institute of Mental Health, 25067 Klecany, Czech Republic; 4Pharmacobiology Department, Center of Research and Advanced Studies, Mexico City 14330, Mexico; lrocha@cinvestav.mx

**Keywords:** neonatal rat, clonazepam, GABA_A_/BZD receptor binding, GABA_B_ receptor binding, subunit mRNA expression

## Abstract

Benzodiazepines (BZDs) are widely used in patients of all ages. Unlike adults, neonatal animals treated with BZDs exhibit a variety of behavioral deficits later in life; however, the mechanisms underlying these deficits are poorly understood. This study aims to examine whether administration of clonazepam (CZP; 1 mg/kg/day) in 7–11-day-old rats affects Gama aminobutyric acid (GABA)ergic receptors in both the short and long terms. Using RT-PCR and quantitative autoradiography, we examined the expression of the selected GABA_A_ receptor subunits (α1, α2, α4, γ2, and δ) and the GABA_B_ B2 subunit, and GABA_A_, benzodiazepine, and GABA_B_ receptor binding 48 h, 1 week, and 2 months after treatment discontinuation. Within one week after CZP cessation, the expression of the α2 subunit was upregulated, whereas that of the δ subunit was downregulated in both the hippocampus and cortex. In the hippocampus, the α4 subunit was downregulated after the 2-month interval. Changes in receptor binding were highly dependent on the receptor type, the interval after treatment cessation, and the brain structure. GABA_A_ receptor binding was increased in almost all of the brain structures after the 48-h interval. BZD-binding was decreased in many brain structures involved in the neuronal networks associated with emotional behavior, anxiety, and cognitive functions after the 2-month interval. Binding of the GABA_B_ receptors changed depending on the interval and brain structure. Overall, the described changes may affect both synaptic development and functioning and may potentially cause behavioral impairment.

## 1. Introduction

Gama aminobutyric acid (GABA) is the main inhibitory neurotransmitter in the adult central nervous system. The role of the GABAergic system in the developing brain appears to differ substantially from that in the adult central nervous system (CNS). During early development, GABA exerts an important neurotrophic function and is implicated in many neurodevelopmental processes (for review, see Reference [1]).

The effects of GABA are mediated by two major types of receptors—the ionotropic GABA_A_ and the metabotropic GABA_B_ receptors. In the adult brain, GABA acts primarily through hyperpolarizing GABA_A_ receptors, ligand-gated Cl^-^ channels. Each receptor comprises three to five different subunits α, β, γ, and δ in heteropentameric structure. Specific GABA_A_ receptor subtypes, defined by the subunit composition, differ by their functional properties, pharmacological sensitivity, location (synaptic vs. extrasynaptic), and distribution in the brain (for review, see Reference [2]). In contrast, metabotropic GABA_B_ receptors are composed of two subunits, GABA_B1_ and GABA_B2_, and they are found both pre- and postsynaptically [3]. The subunit composition, distribution, and properties of GABA receptors at early development differ markedly from those expressed in the adult brain [4].

During early development, GABA is essential for a wide spectrum of developmental phenomena, including cell proliferation and migration, synaptic formation and plasticity, neuronal maturation, and network formation [5]. For this reason, neurodevelopmental disturbances of the GABAergic system have been associated with neuropsychiatric disorders and behavioral dysfunctions. The finding that GABA has critical functions in brain development, in particular during the late embryonic and neonatal period, raises questions regarding the safety and possible undesirable effects of GABAergic drugs that have, for decades, been commonly used as anesthetics, sedatives, and anticonvulsants in patients of all age groups.

Benzodiazepines (BZDs) exert their effects via modulation of the GABA_A_ receptor properties. They have multiple clinical uses, including therapy of anxiety, insomnia, muscle spasm, alcohol withdrawal, and seizures in patients of all age groups—including neonates, children, and pregnant women [6]; thus, the possible undesirable effects of BZDs on the developing brain are of great concern. Studies conducted using laboratory animals have shown that pre- and/or postnatal BZD exposure has short- and long-lasting consequences on brain chemistry and behavior and that it leads to increased apoptosis and suppresses neurogenesis and synaptogenesis (for review, see References [7,8,9,10]). Also, in humans, prenatal exposure to BZDs is associated with increased risk of behavioral problems later in life [11].

Previous studies have demonstrated that the effects of BZDs are stable during development and that these drugs exert strong anticonvulsant and anxiolytic effects even in neonatal animals [12,13]. Neonatal rats also develop side effects similar to those reported in adults, such as sedation [14]. However, unlike in adults, some of the effects of early BZD exposure can outlast the drug itself. Neonatal animals treated for a short period of time with clonazepam (CZP) were shown to exhibit a variety of behavioral deficits during adolescence and adulthood, including deficits in learning, emotions, and social behavior [15,16], and similar alterations were reported also after perinatal exposure to other benzodiazepines [17,18,19,20]. Infantile rats have also been shown to develop signs of BZD withdrawal after very short exposure [12,14,21]. Irrespective of the obvious clinical importance of BZDs, studies regarding the long-term effects of early benzodiazepine exposure are sparse and the mechanisms underlying the associated changes are still poorly described.

In our previous study [22], we described changes in the glutamatergic receptors after short-term administration of CZP to perinatal rats. The expressions of NMDA (N-methyl D aspartate) and AMPA (α-amino-3-hydroxy-5-methyl-4-isoxazolepropionic acid) glutamate receptor subunit mRNAs and NMDA receptor binding were assessed in three intervals after therapy cessation—48 h, 1 week, and 2 months. Using the remaining tissue, the present study aimed to characterize the changes of the GABA_A_ and GABA_B_ receptors and analyzed the expressions of α1, α2, α4 γ2, and δ GABA_A_ receptor subunit mRNA; GABA_B2_ receptor subunit mRNA; and BZD, GABA_A_, and GABA_B_ receptor binding.

## 2. Results

### 2.1. The Effect of CZP Administration on GABA_A_ and GABA_B_ R2 Receptor (Rp) Subunit mRNA Expression 

Early exposure to CZP resulted in moderate changes only in GABA_A_ Rp subunit mRNA expression. Significant differences between the control and CZP-treated animals were found only in short intervals after CZP cessation. In the cortex, early CZP exposure caused a significant decrease of total mRNA for all of the evaluated GABA_A_ Rp subunits 48 h after treatment cessation (*t = 2.526 df = 92; p = 0.0132*). In the hippocampus, total mRNA was not affected at any interval. Interestingly, the expression of GABA_A_ Rp δ subunit mRNA was downregulated in the cortex as well as in the hippocampus two days after the end of therapy (Figure 1A and Table A1). The receptors containing the δ subunit were insensitive to classical BZDs, suggesting that changes in δ subunit expression are not due to a direct interaction between the receptors and CZP. The role of the δ subunit-containing receptors in the regulation of neuronal excitability, synaptic plasticity, and network synchrony [23,24] predetermines their important function in the maturation of brain circuitry. Therefore, even transient alterations of these receptors during the critical developmental period may result in enduring changes in brain structure. 

In addition to the changes of δ subunit expression, early CZP exposure significantly upregulated α2 subunit mRNA expression in the hippocampus 48 h after the end of therapy as well as in the cortex 1 week later (Figure 1A and Table A1). The δ and α2 subunits exhibit opposing developmental profiles; whereas δ subunit mRNA appears gradually during postnatal development with a dramatic increase in expression in the second week after birth, receptors containing the α2 subunit disappear from many areas as maturation progresses and are replaced with the α1 subunit containing assembly [25]. Therefore, the CZP-induced shifts in δ and α2 subunit expressions may be due to developmental delay. Such alterations in GABA_A_ receptor development can affect the formation of brain circuitry and can participate in enduring alterations of the brain’s structure and functions. 

Administration of CZP did not affect the expression of GABA_B_ R2 mRNA receptor subunit at any interval used in this study (Figure 1B and Table A1). The expression of synaptophysin mRNA tended to be reduced by approximately 15% in the hippocampus 2 months after the end of treatment (Figure 1C and Table A1), but difference between the treated animals and controls was not significant. 

### 2.2. Effect of CZP Administration on [3H] Muscimol Binding and [3H] Flunitrazepam Binding 

Our results revealed striking differences in the effects of early CZP exposure on [3H] muscimol and [3H] flunitrazepam binding. Muscimol is regarded as a universal nonselective GABA_A_-site agonist with exceptionally high sensitivity to the δ subunit-containing GABA_A_ receptors [26,27]. On the other hand, [3H] flunitrazepam binds only to a specific, benzodiazepine-sensitive subpopulation of the GABA_A_ receptors. In our study, early CZP exposure resulted in only a transient and short-lasting increase of [3H] muscimol binding (Figure 2 and Table A2). In almost all of the measured brain areas, binding was significantly elevated 48 h after treatment cessation and returned to control levels thereafter. Interestingly, a marked increase of [3H] muscimol binding was not accompanied by a parallel increase of [3H] flunitrazepam binding (Figure 3 and Table A3). In contrast, binding to the BZD receptors tended to be lower in the CZP-exposed animals 48 h after the end of treatment in most of the cortical, amygdalar, and thalamic areas; a significant decrease was, however, found only in the ventromedial thalamus. The discrepancy between [3H] muscimol and [3H] flunitrazepam binding is surprising and is not easy to explain. It may be related to a reduction of the coupling between the GABA site and the BZD site, which has been documented before [28], or by different receptor selectivity of both ligands. 

[3H] muscimol binding quickly returned back to control levels, and no differences between the control and CZP animals were observed at later points (Figure 2 and Table A2). By contrast, alterations in [3H] flunitrazepam binding caused by early CZP exposure were delayed and became evident 2 months after the end of therapy. In most of the cortical, amygdalar, and hippocampal areas measured in our study, [3H] flunitrazepam binding tended to be lower by 20–50% at this interval, but differences between the control and CZP-treated animals reached a significant level only in the medial amygdala, the nucleus (ncl.) accumbens, the dorsal hippocampus, and the periaqeductal grey (Figure 3 and Table A3). All of these structures are part of the neuronal networks associated with emotional behavior, anxiety, and cognitive functions, i.e., with the behavioral domains that are permanently affected by administration of BZDs during critical developmental periods. Our findings support the hypothesis that, in the immature brain, drug-induced changes can be delayed and can become detectable later in life (for review, see Reference [29]). 

Another discrepancy occurred between the downregulated expression of the δ subunit seen 48 h after treatment cessation and an increase in [3H] muscimol binding at the same interval. As mentioned above, muscimol exhibits high sensitivity to the δ subunit-containing GABA_A_ receptors [27]; thus, one can expect a decrease of binding due to reduction of the number of receptors. In our study, we assessed only mRNA expression and not the protein levels, and changes in messages do not necessarily result in changes in protein levels or vice versa. The functional significance of δ subunit mRNA expression must therefore be interpreted with caution.

Panels on the right: Representative autoradiograms illustrating the distribution of binding to GABA_A_ receptors labeled with [3H] muscimol in the brain sections at the level of the striatum 48 h (on the top) and 2 months (on the bottom) after treatment withdrawal of rats treated with vehicle (C—left panel) and clonazepam (CZP—right panels). An increase in binding is obvious in most of the brain structures 48 h after therapy discontinuation; high binding appears as darker areas.

### 2.3. Effect of CZP Administration on [3H] CGP54626 Binding 

The pattern of changes of [3H] CGP54626 binding, used to assess the GABA_B_ receptors, induced by early CZP exposure was highly dependent on brain structure and the interval after CZP cessation (Figure 4 and Table A4). Long-term, early CZP exposure resulted in an increase of [3H] CGP54626 in the sensorimotor cortex, the ventroposterior thalamus, and the medial and basolateral amygdala, whereas in the anterior amygdala, binding was reduced compared to the control. Two days after the end of administration, a significant increase in binding was observed in the central amygdala and the substantia nigra pars compacta, whereas in the ncl. accumbens and the ventrolateral and lateral thalamus, binding was decreased. One week later, an increase in binding occurred in several areas of the thalamus (ventromedial, lateral, and ventrolateral) (Figure 4 and Table A4). The GABA_B_ receptors were critically involved in the regulation of synaptic activity. Although no physical contact or complex formation was reported, there is functional crosstalk between the GABA_B_ receptors and the ionotropic glutamate receptors, with both the NMDA and AMPA receptors involved in GABA_B_ receptor endocytosis, trafficking, degradation, and phosphorylation. As reported before, early CZP exposure leads to both short- and long-term changes of the NMDA and AMPA receptors, which can be partially responsible for the changes observed in the GABA_B_ receptors in our study.

Taken together, the results demonstrate that early CZP exposure leads to both transient and permanent changes in the GABAA and GABAB receptors and, as reported previously, also in the glutamatergic receptors [22]. Alterations of both the GABA and glutamate receptors occur shortly after the end of CZP administration, i.e., during a very sensitive developmental period of brain growth spurt [30,31]. We hypothesize that these changes can underlie disturbances in brain network development and can result in the loss of neurons or synapses [10,32], thus contributing to long-term enduring changes in the structure or number of receptors, which consequently results in behavioral deficits.

## 3. Discussion

Our data demonstrate that exposure to CZP during the early stages of postnatal development affects the BZD/GABA_A_ and GABA_B_ receptors, both in the short and long terms after drug cessation. In adults exposed to CZP during the neonatal period, alterations were seen in BZDs and GABA_B_ binding. The exact pattern of change was dependent on the individual receptor and/or the receptor subunit, brain structure, and interval after CZP discontinuation. 

Interestingly, CZP administration affected the expression of individual GABA_A_ Rp subunit mRNAs only moderately and only within the first week after CZP cessation (Figure 1 and Table A1). In addition, the pattern of these changes differed considerably from those described in adults exposed chronically to BZDs. It has to be emphasized, however, that studies regarding the effects of BZD exposure on GABA_A_ receptor subunit composition performed in adult animals have brought about inconsistent results. The pattern of change seems to be highly dependent on the BZD used; the duration of exposure; the brain structure; and, most significantly, the time of the analysis (during treatment vs. during withdrawal) (for review, see Reference [33]). In our study, early CZP exposure resulted in overexpression of the α2 subunit within one week after treatment cessation, whereas in adult animals, the expression of α2 was not affected by chronic BZD treatment [34,35,36]. Moreover, the effects of long-term BZD administration on the expression of α1 subunit mRNA were significantly different in neonates that from those in adult rats. Depending on the timing of the analysis, α1 subunit mRNA was either upregulated or downregulated during treatment and/or the withdrawal period in adults [37,38,39], whereas in our study, the expression of the α1 subunit was not affected in any posttreatment interval or in any brain structure analyzed. Although GABA is already in abundance in the prenatal brain (for review, see Reference [40]), its function and receptor composition change during development and each receptor subunit exhibits a unique developmental profile and an age-dependent distribution [25,41,42,43,44]. The expression of receptors containing the αl subunit increases markedly throughout most of the brain during postnatal development, whereas receptors containing the α2 subunit disappear from many areas shortly after the onset of αl subunit expression [25]. This switch coincides with synaptogenesis, suggesting that the emergence of α1 subunit-containing receptors parallels the formation of neuronal circuits [44]. The upregulation of the α2 subunit seen shortly after early CZP exposure may reflect a developmental delay in the α2/α1 subunit switch and may affect the development of brain circuitry, as it occurs during a sensitive developmental period. 

Significant downregulation of the δ subunit expression was seen in both the cortex and the hippocampus 48 h after the end of CZP exposure (Figure 1 and Table A1). Activation of receptors containing the δ subunit generates a persistent tonic current that profoundly affects neuronal excitability [45,46]. During early development, tonic inhibition is important in regulating the excitation/inhibition balance during hippocampal maturation [47]. In a study with transgenic mice that lacked δ GABA_A_ Rp, Korpi and collaborators demonstrated the role of this subunit in dentate gyrus neurogenesis, suggesting that even transient alteration in δ subunit expression can affect hippocampal maturation by several mechanisms [48]. 

Taking these data together, we hypothesize that even limited changes in GABA_A_ receptor composition together with altered composition of the NMDA and AMPA receptors [22] can significantly impair synaptogenesis, formation and maturation of brain circuitry, as well as synaptic plasticity later in life. This hypothesis is supported by a previously published study [32] that showed disrupted synaptic development after early exposure to antiepileptic drugs.

Interestingly, CZP exposure resulted in a marked increase of [3H] muscimol 48 h after treatment withdrawal (Figure 2 and Table A2). At the same time point, [3H] flunitrazepam binding was not affected (Figure 3 and Table A3). Thus far, there is very little experimental evidence to demonstrate that long-term BZD administration results in changes of BZD Rp binding, and the results published to date are conflicting (for review, see Reference [49]). Most studies conducted in adults chronically exposed to BZDs found no change in BZD Rp number or a decrease in receptor density after a very high dose of benzodiazepines [50]. An increase of both [3H] muscimol and [3H] flunitrazepam binding after BZD exposure was demonstrated in vitro in cell cultures [51]. However, the mechanisms responsible for the discrepancy seen in our study have to be further studied. 

Whereas [3H] flunitrazepam binding was not affected in the withdrawal period, two months after treatment cessation, a decrease of BZD receptor density was observed in many cortical areas, including in both the dorsal and ventral hippocampus, the ncl. accumbens, the caudate putamen, and the periaqueductal grey (Figure 3 and Table A3). These structures are part of the neuronal networks associated with emotional behavior, anxiety, and cognitive functions (for review, see References [52,53,54,55,56]), which are impaired in animals exposed to BZDs early in life [15,16,17,18,19,20,57]. Whether the decrease of [3H] flunitrazepam binding is associated with a decreased number of neurons or synapses remains to be further studied.

Early exposure to CZP also affected [3H] CGP54626 binding to metabotropic GABA_B_ receptors, and the changes were structure and interval dependent (Figure 4 and Table A4). GABA_B_ receptors are located presynaptically as well as postsynaptically and play an important role in controlling synaptic activity. Released GABA can feed back into the GABA_B_ autoreceptors located on the GABAergic terminals, can activate the GABA_B_ receptors on a neighboring gutamatergic terminal, and can inhibit neurotransmitter release. GABA can also signal postsynaptic GABA_B_ receptors with the potential to modulate various properties of postsynaptic transmission (for review, see Reference [58]). GABA_B_ receptors appear to exhibit neurotrophic properties during development [59]. Data from in vivo studies are inconsistent, but the complete silencing of GABA_B_ receptors in mice causes many behavioral alterations, such as epilepsy and hyperalgesia [60,61]. In utero knockdown of GABA_B_ receptors alters cortical development in rats [62]. Early CZP expression affected GABA_B_ receptor density in a structure- and interval-dependent manner. We presume that the changes seen in GABA_B_ receptor density within the withdrawal period, i.e., during the crucial period of postnatal brain development, can participate in an impairment of the normal development of neural networks, with functional consequences occurring later in life. GABA_B_ receptor activity was found to modulate synaptic plasticity in the adult nervous system as well as LTP and many behavioral functions (for review, see Reference [63]), and GABA_B_ receptors have been implicated in a wide variety of neuropsychiatric disorders (for review, see Reference [64]). Therefore, the chronic changes in the GABA_B_ receptors observed in in our study—in several amygdalar nuclei, in the sensorimotor cortex, and in the ventrolateral thalamus—can directly participate in the functional deficits described before. 

To the best of our knowledge, this and our previous study [22] are the first reports showing the long-term effects of relatively short-lasting BZD exposure during the neonatal period on the GABA_A_, GABA_B_, and glutamate receptors. However, there are several limitations in our studies that have to be noted. First, not all changes found in mRNA expression have to correlate with changes in protein levels. Therefore, any interpretation of our data that suggests changes in receptor composition has to be taken with caution. Second, although our study primarily aimed to map principal receptor changes at three intervals after treatment cessation, the mechanisms underlying the observed changes remain to be clarified. Previously published data suggest, however, several possible mechanisms that can play a role in the observed alterations. Early exposure to BZDs was found to increase apoptosis and to supress neurogenesis in many brain areas [65,66,67], and pro-apoptotic drugs have been found to disrupt synaptic development [32]. All of these mechanisms are likely involved in the receptor alterations seen in both our studies mapping the effect of early CZP on the GABA and glutamate [22] receptors. We also expect that the interplay between the GABAergic and glutamatergic receptors participates substantially in long-term receptor changes because alterations of one receptor type can trigger compensatory changes in other receptor types. Third, our study does not provide any information concerning receptor changes during BZD exposure. All of the described changes were observed either during the withdrawal period or much later in adulthood. 

Nevertheless, our results support the hypothesis that, in the immature brain, drug-induced changes may be incorporated as a permanent developmental alteration of the system (for review, see Reference [29]). In our study, animals were exposed to CZP and CZP withdrawal during a highly vulnerable period of development, covering the period of growth spurt and increased synaptic plasticity [30,31,64], in order to create a synaptic network and to process properly environmental stimuli. Even transient disruption of the balance between the two major neurotransmitter systems that are critically involved in synaptogenesis and in formation and maturation of neural networks may be responsible for the pathological consequences seen in our previous studies and studies by others. This should be taken into consideration in the development of new and safe drugs for pediatric patients.

## 4. Materials and Methods 

The experiments were performed using male Wistar albino rats (*n* = 90). The day of birth was counted as zero (P0). Rats were housed in a controlled environment (temperature 22 ± 1 °C, humidity 50–60%, lights on 600–1800 h) with free access to food and water. Animals were weaned at P28. All procedures involving animals and their care were conducted according to the ARRIVE (Animal Research: Reporting In Vivo Experiments) guidelines in compliance with national (Act No 246/1992 Coll.) and international laws and policies (EEC Council Directive 86/609, OJ L 358, 1, December 12, 1987; Guide for the Care and Use of Laboratory Animals, U.S. National Research Council, 1996). The Ethical Committee of the Czech Academy of Sciences approved the experimental protocol (Approval No. 128/2013, approval date: 23 September 2013).

### 4.1. Pharmacological Treatment

Clonazepam (CZP) was suspended in physiological saline with addition of Tween 80 (1 mg/5 mL of saline with one drop of Tween 80) and injected intraperitoneally in a dose of 1 mg/kg/day for 5 consecutive days starting at postnatal day 7 (P7) until P11. The selection of the used dose and the duration of administration were done according to our previous studies that demonstrated long-lasting behavioral alterations [15,16] in animals treated in the same way. Control animals received solvent instead. After injection, pups were immediately returned to their dams. Separation from mothers during drug administration never exceeded 20 min. During the drug administration, pups were kept on an electric heating pad connected to a digital thermometer at 34 ± 1 °C to compensate for the immature thermoregulation at this age [68]. 

Body weight was checked daily during drug administration until the end of the experiment. The difference in body weight between two consecutive days was used to assess weight gain. 

### 4.2. Quantitative Real-Time RT-PCR

Hippocampi and sensorimotor cortices obtained from 10 animals per treatment and interval groups were immediately dissected and homogenized in RNAzol RT (Molecular Research Center). Total RNA was extracted by Direct-zol™ RNA MiniPrep (Zymo research, Irvine, CA, USA) according to the manufacturer’s instructions. Total RNA (1 µg) was converted to cDNA using the one-step SuperScript^®^ VILO cDNA Synthesis Kit and Master Mix (Invitrogen, Carlsbad, CA, USA) according to the manufacturer’s instructions. Samples of cDNA (1 μL) were amplified in 20 μL of PCR reaction mixture containing 5× HOT FIREPol^®^ Probe qPCR Mix Plus (Baria, Prague, Czech Repubic) plus TaqMan probes (Life technologies, Carlsbad, CA, USA; Table 1). All qPCR reactions were performed in duplicate in a LightCycler^®^ 480 Instrument (Roche Life Science, Indianapolis, IN, USA) using the following temperature profile: initial denaturation at 95 °C for 15 min, followed by 60 cycles consisting of denaturation at 95 °C for 18 s and annealing/elongation at 60 °C for 60 s. The mean of the crossing point (Cp) obtained from qPCR was normalized to the level of the housekeeping gene Ppia (Cyclophilin A) and then used for analysis of relative gene expression by the ΔΔCT method [69] as described in Kubová et al. [22]. Mean values of the controls were normalized to zero, and values of the CZP-treated animals were plotted as percent difference from the controls (i.e., zero). For statistics, all of the values for the controls were counted as percent distribution around the mean and compared with the treatment groups. The reproducibility of the assays was evaluated using calculation of the coefficient of variation according to following formula: % CV = σ/µ (σ = SD and µ = mean value). The chosen criterion was %CV < 10.

### 4.3. Receptor Binding

Rats (5 animals per treatment and interval group) were sacrificed under light ether anesthesia by decapitation, and the brains were rapidly frozen in pulverized dry ice and stored at –70 °C until processing. Serial sections (1 of 5) through the entire brain were thaw-mounted on gelatin-coated slides and stored at –70 °C until the day of incubation. Serial and parallel sections were produced from each brain for subsequent autoradiography procedures. The brains of the CZP-treated and age-matched control rats were always obtained and examined simultaneously. 

Quantitative autoradiography: The experiments were performed as described previously [70]. Brain sections were removed from the freezer, dried in a stream of cool air, and immediately washed to eliminate endogenous ligands. Then, sections were incubated in a solution with the specific ligand labeled with tritium ([3H]) in the presence or absence of a non-labeled specific ligand. Specific conditions for individual ligands are summarized in Table 2. The specific binding was established from the difference of values between both experimental conditions. Incubation was concluded with two consecutive washes in buffer solution and finally with cold distilled water for 2 s. Sections were quickly dried in a mild steam of cold air. Thereafter, they were arranged in X-ray cassettes with 3H standards (Amersham) and exposed to [3H]-sensitive film (Kodak MR) at 22 °C. Each film allowed the simultaneous exposition of 21 slides plus one standard, i.e., each film included sections from 7 animals. Each film contained sections from CZP animals and age-matched controls. All slides were processed in one autoradiography assay in order to avoid variability of the experimental conditions.

The film was developed at 18–20 °C using the Kodak D19 developer and fast fixer solutions. In every animal, the optical density was assessed as a mean of 10 measurements that were done in at least three parallel sections for each measured structure. The mean value was calculated and used for statistical evaluation. Optical densities were determined using JAVA Jandel image analysis software. We used tritium standards previously calibrated to brain homogenates with known protein concentrations to allow a transformation of gray values into total binding per milligram of protein as follows: (a) The optical density readings of the standards were used to construct a standard curve to determine tissue radioactivity values for the accompanying tissue sections (dpm/mm^2^). The optical density readings of the standards were used to construct a standard curve to determine tissue radioactivity values for the accompanying tissue sections (dpm/mm^2^). (b) Then, these values (dpm/mm^2^) were converted to fmol/mg of protein based on the specific activity of each [3H] ligand and tissue thickness (20 μm).

### 4.4. Statistics

Sample size was determined in advance based on previous experience and following the principles of the three Rs (replacement, reduction, and refinement; https://www.nc3rs.org.uk/the-3rs). At the beginning of the experiment, individual animals were randomly allocated to a particular treatment group. 

All efforts were made to minimize the number of animals used and their suffering. Before the beginning of the experiment, a simple randomization was used to assign each rat to a particular treatment group. Data analysis was done blind to the treatment. The ages and time points for each group consisted of five to seven animals for the binding study and 10 animals per group used for real-time PCR. Data were analyzed using GraphPad Prism 7 (GraphPad Software, United States) software. Using the D’Agostino–Pearson normality test, all data sets were first analyzed to determine whether the values were derived from a Gaussian distribution. Outliers were identified with the ROUT (Robust regression and Outlier removal) test (Q = 1%). The differences between the age-matched controls and the CZP-treated animals were analyzed using ordinary one-way ANOVA with post hoc Sidak’s multiple comparison test, and a *p*-value < 0.05 was required for significance. Data are expressed as the means ± SEM and plotted as % of controls.

## 5. Conclusions

Our results support the hypothesis that, in the immature brain, drug-induced changes may be incorporated as a permanent developmental alteration of the system (for review, see Reference [29]). In our study, animals were exposed to CZP and CZP withdrawal during a highly vulnerable period of development, covering the period of growth spurt and increased synaptic plasticity [30,31,64], in order to create a synaptic network and to process properly environmental stimuli. Even transient disruption of the balance between the two major neurotransmitter systems—glutamatergic and GABAergic—that are critically involved in the synaptogenesis and in formation and maturation of neural networks may be responsible for the pathological consequences seen in our previous studies and studies by others. This should be taken into consideration in the development of new and safe drugs for pediatric patients.

## Figures and Tables

**Figure 1 ijms-21-03184-f001:**
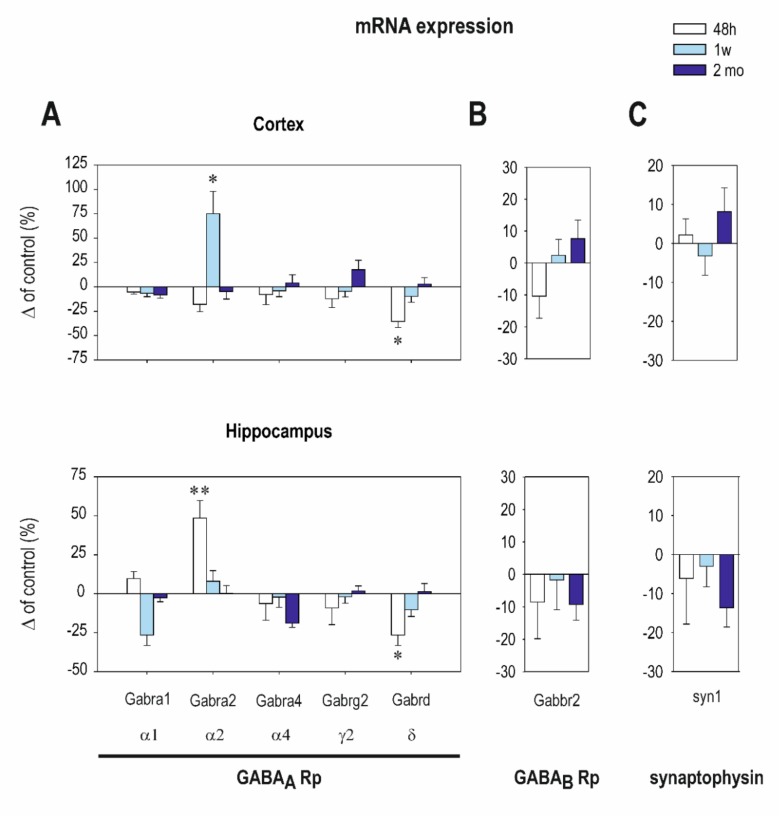
The transcription levels of the Gama aminobutyric acid A (GABA_A_) receptor subunits (**A**) and the GABA_B_ receptor subunit (**B**) in the cortex (on the top) and in the hippocampus (on the bottom) at three intervals after clonazepam (CZP) cessation (48 h, 1 week, and 2 months—legend at the top). (**C**) The graphs on the right demonstrate the transcription levels of synaptophysin at the same intervals. The mRNA levels were determined using quantitative RT-PCR, and values were converted to a percentage of the control values considered as the baseline (zero) levels. Each experimental group consisted of 10 animals. Data were analyzed using GraphPad Prism 7 (GraphPad Software, United States) software. Using the D’Agostino–Pearson normality test, all data sets were first analyzed to determine whether the values were derived from a Gaussian distribution. Outliers were identified with the ROUT test (Q = 1%). The differences between age-matched controls and CZP-treated animals were analyzed using ordinary one-way ANOVA with post hoc Sidak’s multiple comparison test, and a *p*-value < 0.05 was required for significance. Data are expressed as the means ± SEM and plotted as a % of the controls. Asterisks denote significant differences from the controls (** p < 0.05; ** p < 0.01*).

**Figure 2 ijms-21-03184-f002:**
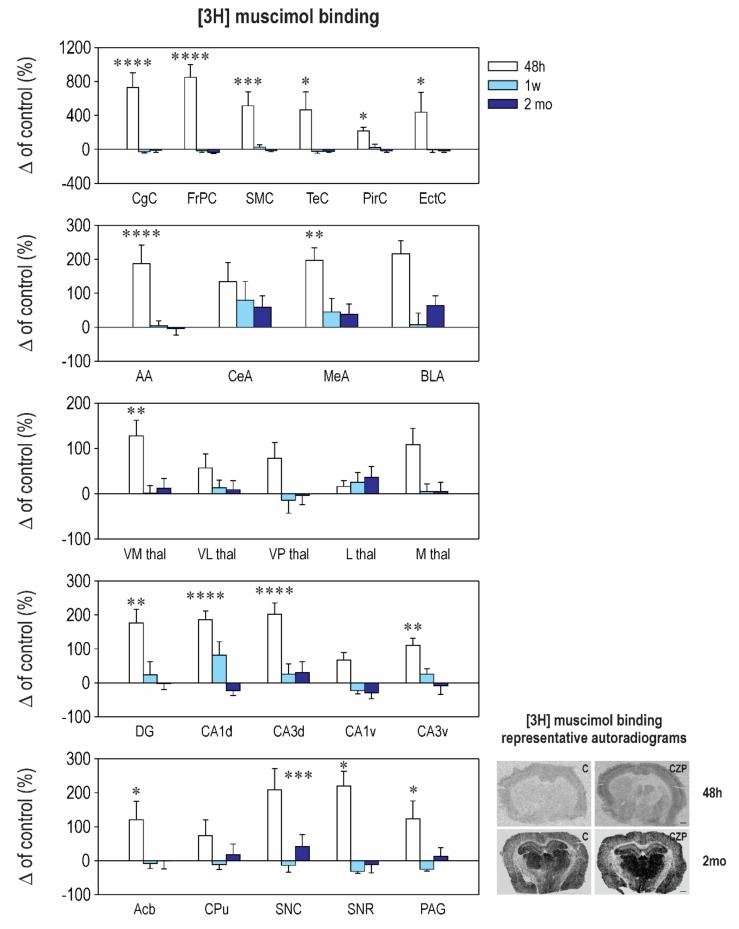
Relative changes in [3H] muscimol binding to GABA_A_ receptors (means ± SEM): The binding in the control animals is considered as the baseline (zero) level. From the top to the bottom: cortical structures, amygdalar structures, thalamic structures, and hippocampal structures. On the bottom: Ncl. accumbens (Acb), caudate putamen (CPu), substantia nigra pars compacta (SNC), substantia nigra pars reticulate (SNR), and periaqueductal gray (PAG). [3H] Muscimol binding was assessed at three different intervals (48 h, 1 week, and 2 months—legend at the top) after CZP cessation. Abbreviations: CgC, cingulate cortex; FrPc, frontoparietal cortex; SMC, sensorimotor cortex; TeC, temporal cortex; PirC, piriform cortex; EctC, entorhinal cortex; AA, anterior amygdala; CeA, central amygdala; MeA, medial amygdala; BLA, basolateral amygdala; VN thal, ventromedial thalamus; VL thal, ventrolateral thalamus; VP thal, ventroposterior thalamus; L thal, lateral thalamus; M thal, medial thalamus; DG-, dentate gyrus of the hippocampus; CA1, CA1 subfield of the hippocampus; CA3, CA3 subfield of the hippocampus; d, dorsal; v, ventral. Each experimental group consisted of 5 animals. Asterisks denote significant differences from the controls (** p < 0.05; ** p < 0.01; *** p < 0.001; **** p < 0.0001*). Scale bar = 1 mm. For other details, see Figure 1.

**Figure 3 ijms-21-03184-f003:**
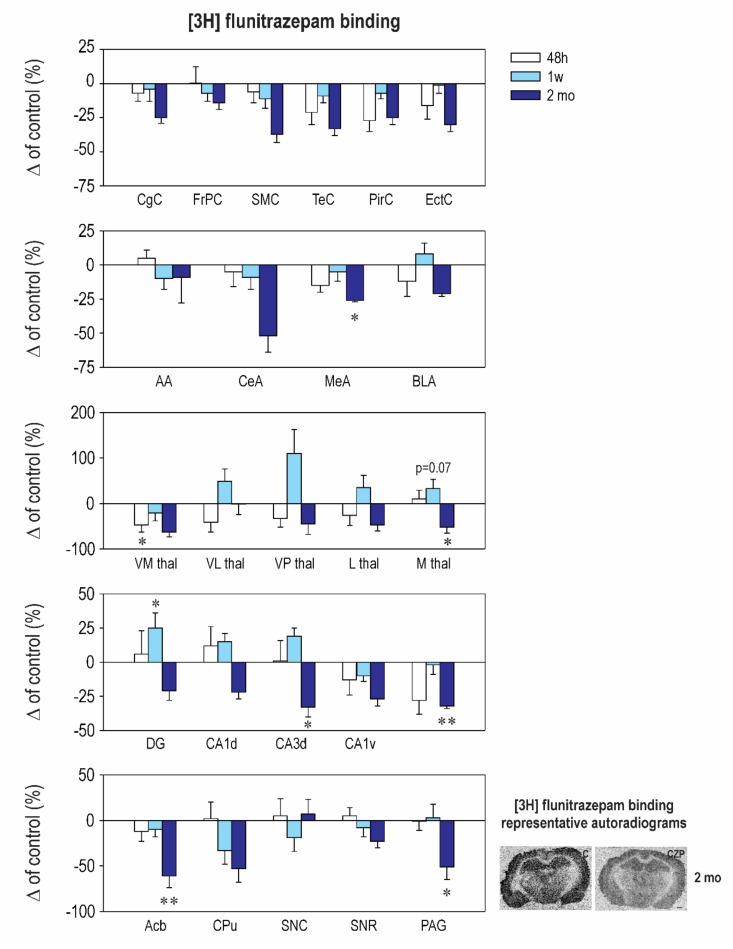
Relative changes in [3H] flunitrazepam binding to Benzodiazepine (BZD) receptors (means ± SEM). Panels on the right: Representative autoradiograms showing decreased binding to BZD Rp labeled with [3H] flunitrazepam. Sections were taken at the level of the hippocampus 2 months after the end of therapy. Details as in Figure 1 and Figure 2. Asterisks denote significant differences from the controls (** p < 0.05; ** p < 0.01*). Scale bar = 1 mm.

**Figure 4 ijms-21-03184-f004:**
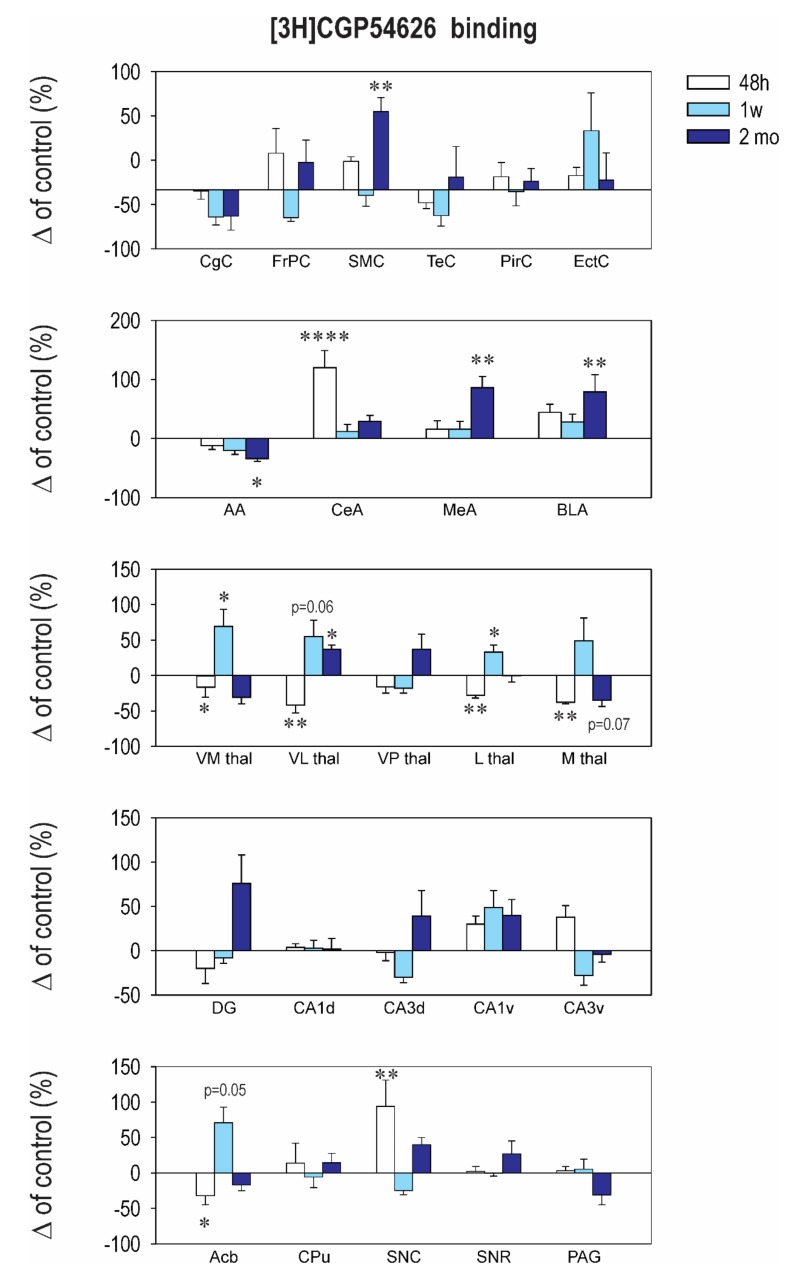
Relative changes in [3H] CGP54626 binding to the GABA_B_ receptors. Details as in Figure 1 and Figure 2. Asterisks denote significant differences from the controls (** p < 0.05; ** p < 0.01; **** p < 0.0001*).

**Table 1 ijms-21-03184-t001:** List of TaqMan probes used.

Ref. No	Gene Symbol	Gene Name	Gene Aliases
Rn00690933_m1	Ppia	peptidylprolyl isomerase A, cyclophilin A	CYCA, CyP-A
Rn00788315_m1	Gabra1	gamma-aminobutyric acid (GABA) A receptor, alpha 1	-
Rn01413643_m1	Gabra2	gamma-aminobutyric acid (GABA) A receptor, alpha 2	-
Rn00589846_m1	Gabra4	gamma-aminobutyric acid (GABA) A receptor, alpha 4	-
Rn01464079_m1	Gabrg2	gamma-aminobutyric acid (GABA) A receptor, gamma 2	-
Rn00568740_m1	Gabrd	gamma-aminobutyric acid (GABA) A receptor, delta	GABAA-RD
Rn00561986_m1	Syp	synaptophysin	Syp1
Rn00582550_m1	Gabbr2	gamma-aminobutyric acid (GABA) B receptor 2	Gpr51

**Table 2 ijms-21-03184-t002:** Conditions for the autoradiography experiments.

Binding	Ligand (nM) and S.A.	Buffer pH 7.4	Incubation	Exposition (RT)	Non-Labeled Ligand
GABA_A_	[3H] Muscimol (20 nM); 20 Ci/mmol	Tris citrate (50 mM)	45 min at 4 °C	8 weeks	GABA (10 µM)
GABA_B_	[3H] CGP54626 (4 nM); 30 Ci/mmol	Tris HCl (50 mM) and CaCl_2_ (10 mM)	90 min at 22 °C	12 weeks	CGP 55845 (100 µM)
BDZ	[3H] Flunirazepam (2 nM); 85.2 Ci/mmol	Tris HCl (170 mM)	45 min at 4 °C	3 weeks	Clonazepam (1µM)

Abbreviations: S.A., specific activity; RT, room temperature.

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
