# Peer review of "Neonatal Clonazepam Administration Induced Long-Lasting Changes in GABAA and GABAB Receptors"

_ijms, 2020, doi:10.3390/ijms21093184_

Round 1

Reviewer 1 Report

In their manuscript entitled “Neonatal clonazepam administration induced long-lasting changes in GABAA and GABAB receptors” Hana Kubová and coworkers investigated the changes in the mRNA expression levels of selected GABAreceptor subunits as well as in muscimol or Flunitrazepam binding after early postnatal clonazepam treatment. The topic of this study is interesting and the experiments are well conducted. However, the statistical analysis of this study must be improved and there are some flaws in the introduction and discussion of the experiments. In addition, the manuscript need mild language editing.

Major points:

Line 62: Regarding this issue you cite 3 review articles from the 80th, whereas there is compelling evidence on this topic published in the meantime, including studies in humans. Thus I strongly recommend to update this part of the introduction.

Line 366: Statistics: As you perform multiple comparisons for the analysis of your experiments, you should correct the t-test for the bias induced by this approach (e.g. Bonferroni-Holm).

Discussion: In general, I missed a discussion of the mechanisms and the functional implications of the observed changes. E.g. is the reduced synaptophysin mRNA expression due by a reduced number of neurons (e.g. reports from Ikonomidou lab), synapses (e.g. Huang lab) or does it even reflect more stable synaptic terminals (e.g. Fritschy lab)? Are the reduced levels of GABA receptors caused primarily by a direct effect on the GABA transmission of do they result from a global reduction in neuronal activity under benzodiazepine treatment? How do your findings support or contradict observations in human patients / experimental animals who experiences prenatal exposure?

Minor:

Line 19: Introduce abbreviation “GABAARp” already here or don’t used abbreviation in the abstract (recommended).

Line 48: “During early development, GABA is depolarizing and mostly excitatory”. Please tone down this statement, as recent in-vivo studies questioned whether GABA is really excitatory in early postnatal brain, at least in the neocortex of newborn rodents.

Line 85: Why here df = 1548?

Fig. 1: I just wonder why the order cortex –hippocampus in the figure was different from the text order.

Legend to Fig. 1: Please note the wrong reference of the synaptophysin plots (B in the figure) and the GABA(B) plots (C in figure) in the legend.

Line 138: In my opinion the terminology GABA(A) Rp binding and BZD Rp Binding is misleading, as the “benzodiazepine receptor” is, as molecular target, also a subset of GABA(A) receptors. Thus I would recommend to strictly stick to the descriptive terms “muscimol binding” and “BZD binding”.

Figures: I suggest to order the references/numbering of the figures. It is misleading that Fig. 4 appears after Fig. 3, although Fig. 4 illustrates the results of Fig. 2

Line 295: Please clarify whether you controlled the body temperature of the pups (rectal thermometer, then 34°C s rather low) or whether you controlled the temperature of the environment with the heating pad.

Line 300: Can you also provide the number of used animals for the q-RT-PCR experiments already here (like you did for the receptor binding assays). However, I would suggest that you provide the number of experimental animals more prominent in the results part and/or the figure legends, as this value helps to estimate the statistical power of our results.

Finally, the manuscript requires small language editing. Please find a few suggestions here:

Line 27: “… in almost all brain structures including (?) neuronal networks associated …”

Line 38 “.. is implicated… ”

Line 56: BZDs doesn’t modulate the neurotransmitter GABA but they modulate the properties of the GABAA receptor. Please rephrase.

Line 67 “for a short period ….. exhibited a variety …”

Line 102: “Synaptophysin”

Line 108 “In all measured structures”

Author Response

Referee 1

We would like to express our thanks for referee’s comments and valuable suggestions. We hope that manuscript was improved and mistakes were corrected.

  • Line 66 -68 More recent articles were added in Introduction and we also added clinical study on impact of prenatal BZDs exposure on behavioural problems in children. Authors however would like to point out that originally used articles from 80th are still the most comprehensive studies published on this topic
  • Line 365 We controlled temperature of the heating pad, not core temperature. Tis was corrected in the text.
  • Line 434-443. Data were re-analysed using ordinary one-way ANOVA with post-hoc Sidak’s multiple comparison Different statistical analysis brought some differences in results and as requested by Referee 2, part of manuscript describing results was reformulated. Figures and tables were re-done and all mistakes, pointed out by referees were corrected.
  • We extended discussion and we extended data interpretation part. Possible role of BZD effects on neurogenesis and synaptogenesis as well as role of BZD-induced enhancement of apoptosis is discussed and several new references were added (line 281-282, 327-330) It has to be noted that new statistical analysis not confirm significant difference in synaptophysin mRNA expression between controls and CZP exposed
  • Line 52-54. Sentence “During early development, GABA is depolarizing and mostly excitatory…” was replaced by sentence During early development, GABA is essential for wide spectrum of developmental phenomena including cell proliferation and migration, synaptic formation and plasticity, neuronal maturation and network formation [Wang and Kriegstein, 2009]. Whole paragraph was
  • Number of animals used for q-RT-PCR experiments was included in Methods (line 371) and in figure legend. We apologize for omitting this important information in original version of our manuscript.
  • Manuscript underwent professional English editing as requested by editor and both referees.

Reviewer 2 Report

The manuscript by Hana et al. describes the short- and long-term effects of GABAergic receptors on transient administration of clonazepam to neonatal rats. Using RT-PCR and classic quantitative autoradiography, the experiments are straightforward and noting all that special, but the data probably is useful for further cellular research for the neuronal network formation under the dosage. In this manuscript, the authors wrote well in the Discussion part, but the Results part is too short. The order and presentation of some figures are not suitable in the paper.

Major comments:
The results part is difficult to read, and the current contents are unacceptable for the below reasons.
1) Some quantitative data in Fig 2 and 3 were acquired using autoradiogram images in Fig 4. The order of these figs is not suitable. The authors should insert some representative autoradiograms they used for the calculation into Fig 2 and 3.
2) No interpretations and experimental conclusions in the results part. Please add these descriptions, and reconsider moving some sentences from the discussion part to the results.
3) Each number of figures or tables should insert to the corresponding sentences, not subtitles in the results.
4) the many values of t, df, and p should remove and summarise to a supplementary table.

Minor comment:
Please insert figure numbers also in the corresponding sentences in the discussion part.

Author Response

Referee 2

Authors thank to Referee 2 for recommendation that helped us to improve our manuscript.

  • As requested, Results were completely re-written. We included brief data interpretation and conclusion into each result part and we believe that now this part is more “reader friendly” and easier to read. We also moved all statistical details into tables 1 – 4. According to request of Referee 1 data were re-analyzed using ordinary one-way ANOVA with post-hoc Sidak’s multiple comparison test, therefore there are some differences compared to the 1st version of our manuscript.
  • We added numbers of figures and tables into text of Results and in Discussion
  • Manuscript underwent professional English editing as requested by editor and both referees.

Round 2

Reviewer 2 Report

The authors have revised the manuscript according to my comments. I have not further comments.